# Super-resolution techniques to simulate electronic spectra of large molecular systems

Matthias Kick [1] ✉, Ezra Alexander[1], Anton Beiersdorfer[2] & Troy Van Voorhis [1]

An accurate treatment of electronic spectra in large systems with a technique such as time-dependent density functional theory is computationally challenging. Due to the Nyquist sampling theorem, direct real-time simulations must be prohibitively long to achieve suitably sharp resolution in frequency space. Super-resolution techniques such as compressed sensing and MUSIC assume only a small number of excitations contribute to the spectrum, which fails in large molecular systems where the number of excitations is typically very large. We present an approach that combines exact short-time dynamics with approximate frequency space methods to capture large narrow features embedded in a dense manifold of smaller nearby peaks. We show that our approach can accurately capture narrow features and a broad quasi-continuum of states simultaneously, even when the features overlap in frequency. Our approach is able to reduce the required simulation time to achieve reasonable accuracy by a factor of 20-40 with respect to standard Fourier analysis and shows promise for accurately predicting the whole spectrum of large molecules and materials.

Electronic excitations in molecules and materials are important for understanding various kinds of phenomena such as photo-excitation in solar cells, optical excitations in OLEDS and quantum dots[1–11]. Theoretically, electronic excitations can be obtained by analysing the frequency components of the time-dependent dipole moment obtained from real-time propagation[12]. Among various other methods such GW/BSE[13], EOM[14–16] or ADC[17], time-dependent density functional theory (RT-TDDFT)[12] is the most promising method to calculate the whole spectrum of large systems due to its superior scaling with respect to system size compared to other methods. Because of the computational complexity of real-time simulations for large molecules and materials, one is typically restricted to fairly short time dynamics (e.g. tens of fs). Due to the Nyquist sampling theorem, discrete Fourier analysis of the short-time dynamics fails to capture the narrow features that are critical fingerprints of molecular spectra. Meanwhile, standard super-resolution methods - such as compressed sensing (CS)[18–20], MUSIC[21] and orthogonal matching pursuit[22,23] - typically fail for large molecular systems because they require the number of narrow features to be small, whereas the spectra of large molecules tends to be

quite densely populated. Similarly, linear response approaches like the Casida[24] or Sternheimer equation[25] typically require one-at-a-time identification of roots and likewise fail when the number of desired roots is very large. In this paper we show how exact short-time dynamics can be combined with approximate frequency space results to accurately capture narrow features and a quasi-continuum of states in large molecular systems.

Our approach (BYND—Broad Yet Narrow Description) is illustrated in Fig. 1, for the case of a molecular chromophore adsorbed on a surface of a semiconductor nanocrystal. In this case, a super-resolution method only captures a small number of peaks in the overall spectrum, while discrete Fourier Transform (FT) of the short-time signal recovers only a broad quasi-continuum. In our approach, one first obtains an approximate spectrum - in this case using small matrix approximation (SMA)[26] – that has the right number of peaks in roughly the right locations. Next, the most important narrow features in the spectrum are optimized to match the short-time dynamics. Finally, linear prediction is used to match the intensities of the approximate and optimized spectral features - exactly recovering the short-time

[1]Department of Chemistry, Massachusetts Institute of Technology, Cambridge, MA, USA. [2]Technical University of Munich, Garching, Germany.
✉e-mail: mkick@mit.edu

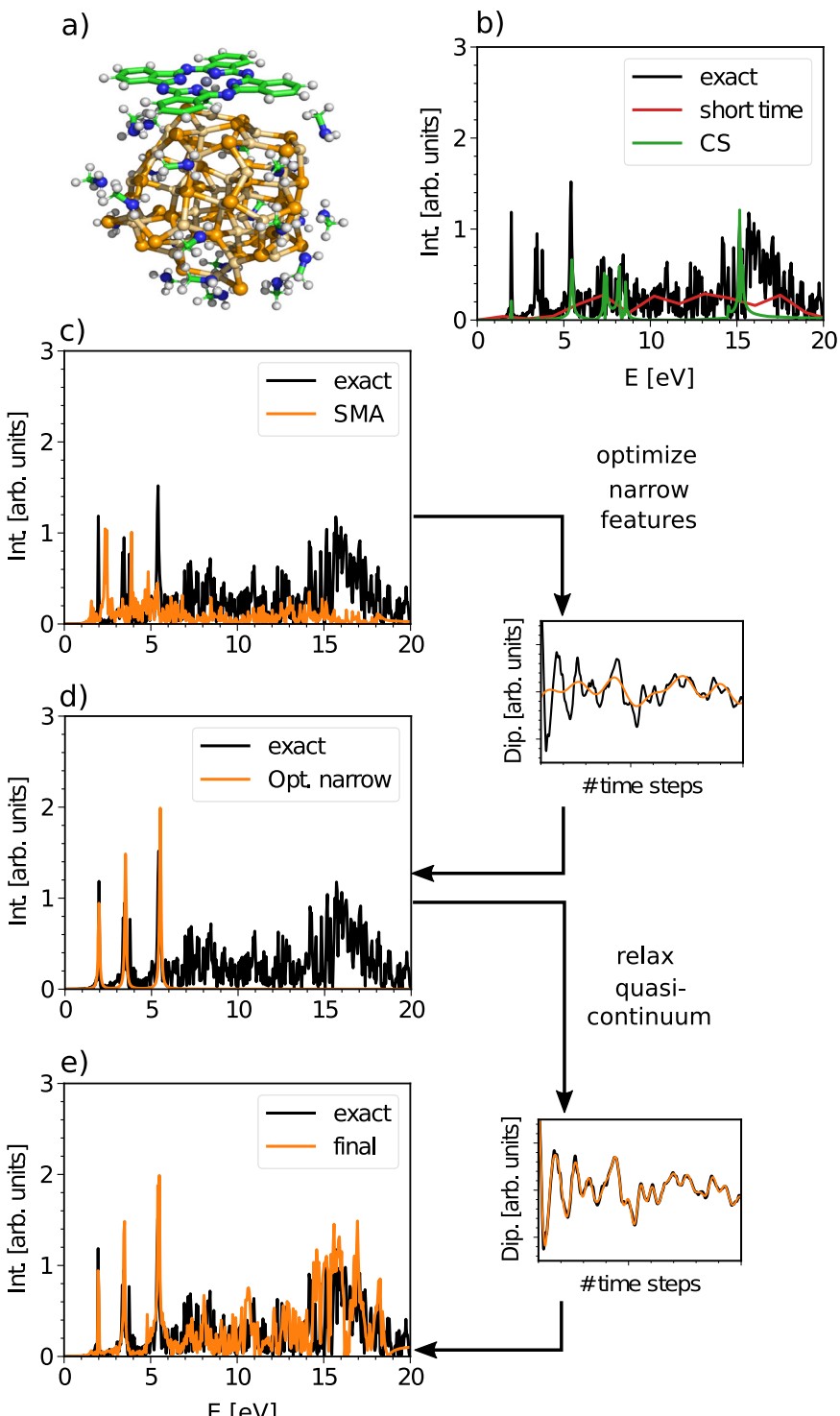

**Fig. 1 | Working principle of BYND. a** Structure of $Cd_{38}Se_{38}+ZnPc+32(NH_2CH_3)$ with in total 301 atoms and 3842 electrons. **b** Comparison of exact time-dependent density functional theory (RT-TDDFT) results (20,000 time steps) with compressed sensing (CS) and a short-time simulation with both 1000 time steps. **c** Approximated spectrum which is calculated using the small matrix approximation (SMA) and which serves as input for BYND. **d** Non-linear optimization is used to locate the narrow features. **e** Final spectrum after linear prediction of the quasi-continuum. Int. refers to the intensity and Dip. refers to the time-dependent electric dipole signal.

signal and yielding a spectrum that is substantially more accurate than CS or discrete FT alone can provide.

BYND successfully finds electronic excitations for large molecular systems where CS and other algorithms fail due to the presence of a quasi-continuum. For our largest test systems, we see standard mean errors between 0.01 and 0.14 eV in narrow feature position with

respect to reference long-time RT-TDDFT. Considering the typical error of TDDFT with respect to experiment is around 0.25 eV[27], our method yields high quality results consistent with standard theoretical practice, useful in interpreting experimental results. Further, we see a reduction in the required computational time between 20- and 40-fold compared to standard FT due to the smaller number of time steps

required by BYND. Thus, BYND enables the simulation of large molecular systems which otherwise would be computationally prohibitive even on modern computer hardware.

This article is structured as follows. We first briefly introduce the theory behind frequency-resolved approximations and exact short-time dynamics. We then move forward to a step-by-step explanation of the working equations of our method. We discuss the performance of BYND on a challenging set of large systems and conclude by discussing future directions for the method.

## Results and discussion
### Linear prediction
Modeling entire spectra from time-dependent signals can, in principle, be achieved by linear prediction[28–30]. The basic idea is that one can predict spectral features from linear combinations of past output values. One simply determines all relevant model parameters directly from the short-time signal[31]. There are techniques which achieve this in time or frequency domain and in principle, if the number of samples is sufficient and if the distance between time steps is adequate, linear prediction is able to model the spectrum with good accuracy[32,33]. However, for the system sizes we are aiming for, sampling enough time steps is computationally prohibitive. Further, if we want to model a spectrum, using linear prediction only, one usually needs an idea of how many frequencies there are and where they are located[32,33]. Even if we would have this information available, the number of frequencies usually exceeds the number of data points by a large amount resulting in an under-determined system which makes it nearly impossible to extract meaningful spectra (see Fig. 2). We discuss these problems in more detail later on in this article 2.4 where we also provide examples.

### SMA
The input required for BYND is an approximate excited state spectrum which shows the right number of peaks at approximately the right energies. To this end, we approximate the pseudo eigenvalue problem

of the Casida equations[24] by employing the SMA. In this approximation, the electronic excitation energies (in frequency space) are given by a simple analytical expression, allowing one to obtain a large number of excitation energies without directly solving the extremely costly pseudo eigenvalue problem. We have implemented the SMA within the FHIaims infrastructure. This implementation allows the rapid evaluation of several thousand exited states easily in systems containing more than 1000 atoms (to be discussed elsewhere).

### RT-TDDFT
Our goal is to improve the frequency information of the SMA by combining it with exact short-time dynamics from a real-time TDDFT (RT-TDDFT) simulation. In RT-TDDFT, the time-dependent Kohn-Sham states are explicitly propagated in time under the influence of an electric field ($E_\lambda$), which usually has the form of a sharp $\delta$-pulse[34,35]. The effect of the electric field pulse is the excitation of all possible electronic excitation modes. Thus, the oscillation of the time-dependent dipole moment from the real-time propagation can be directly linked to the excitation energies of the system. It should be emphasized that BYND can be used with any real-time propagation method. However, due to its superior scaling with respect to system size, real-time TDDFT is the clear choice over other electronic structure methods as we attempt to push toward larger systems. In fact, real-time TDDFT is already widely employed to capture electron dynamics in intermediately-sized molecular and solid-state systems[36–42].

Throughout the text, $\lambda$ and $\mu$ will indicate the direction of the electric field and observed time-dependent dipole moment respectively. For a full optical excitation spectrum one needs to perform three propagations with different orientations of $E_\lambda$ (x, y and z).

### Combining SMA with RT-TDDFT
In this section we will introduce how our method is able to capture both narrow features and the quasi-continuum by combining approximate frequency results from the SMA with short-time RT-

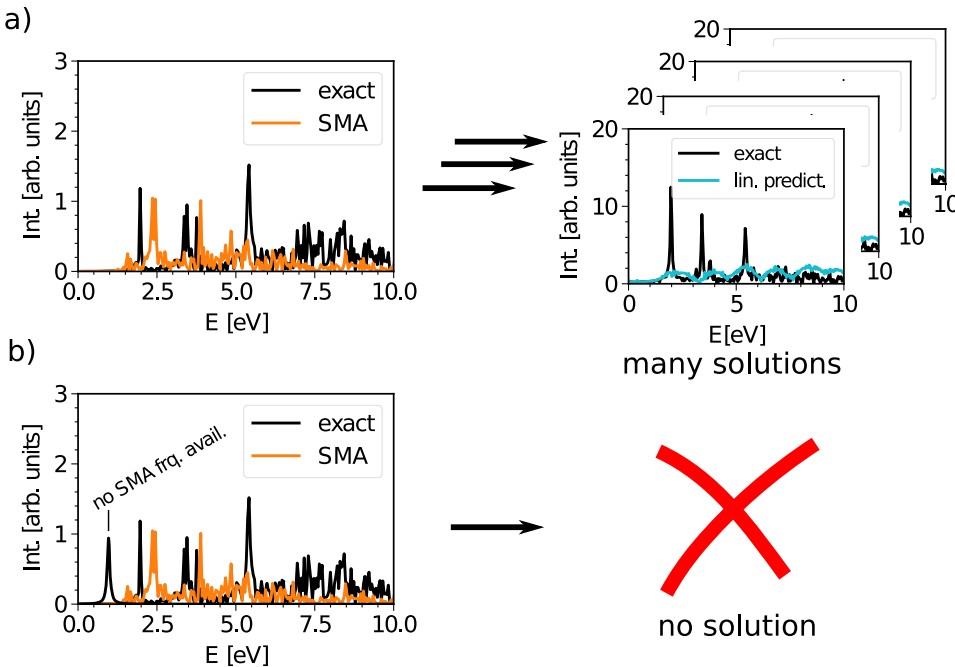

**Fig. 2 | Linear prediction of the excitation dipole spectra of Cd₃₈Se₃₈-ZnPc-32(NH₂CH₃). a** The short-time signal is accurately reproduced in each case, however, narrow features are completely absent in the resulting spectrum. We used a time signal with 1000 time steps and small matrix approximation (SMA) frequencies. To determine the model parameters (amplitudes) for the SMA

frequencies we make use of equation (2). **b** We show an artificial generated spectrum where the first black exact result has no SMA frequency at the corresponding energy. In this case, linear prediction is not able to capture this feature. Intensity is abbreviated with Int.

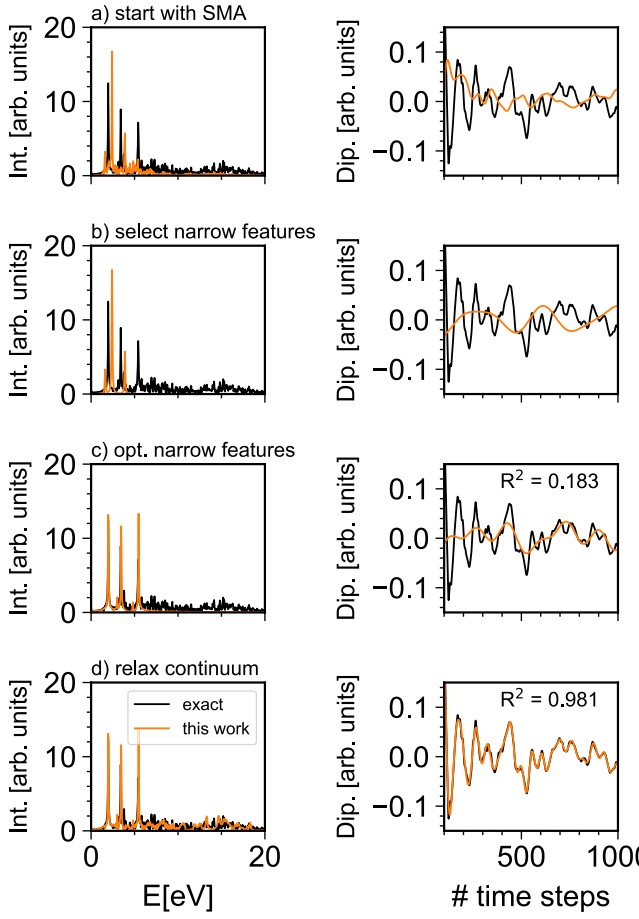

**Fig. 3 | Fourier transform of the time-dependent dipole moment of Cd$_{38}$Se$_{38}$-ZnPc-32(NH$_2$CH$_3$). a** Dipole spectrum obtained from the small matrix approximation (SMA) with the corresponding time-dependent dipole signal (Dip.). **b** Selected narrow features from the SMA calculation. **c** Narrow feature position after optimization. The error with respect to exact short-time dynamics is notably reduced. **d** The full time-dependent signal is now reproduced with high accuracy and all features are correctly reproduced in the spectrum. The SMA signal in (**a**) and (**b**) was scaled to match the maximum amplitude of the time-dependent density functional theory (RT-TDDFT) reference within the given time window. This is only done for a better comparison of the signal wave forms. For the sake of simplicity and without loss of generality, we only show the dipole moment in x-direction after an electric field pulse in the same direction. Int. is the abbreviation for intensity.

TDDFT data. In order to illustrate each step of our approach, the excitation spectrum of Cd$_{38}$Se$_{38}$-ZnPc-32(NH$_2$CH$_3$) will serve as a prototype (Fig. 1). Within this system, bulk, surface and molecular states can easily mix and the excited states of the system blur into a quasi-continuum, requiring the evaluation of a large number of excited states in a small energy window. Specifically, in our example one needs to evaluate roughly 47,000 excited states in order to calculate the spectrum up to an excitation energy of 10 eV. With 3482 total electrons, this system is highly challenging for standard TDDFT and is thus an ideal test of our approach (Fig. 3). The CdSe nanocrystal has been extended into a test set of signals representative of a broad range of common large systems through the addition of aromatic molecules, which add narrow features, and through increasing the size of the nanocrystal, which enhances the continuum region. While these systems are an excellent test bed to study convergence and accuracy of signals with challenging wave forms, additional test systems such as dye-sensitized solar cells, surfaces slabs, molecular aggregates and nano tubes will also be used here to further illustrate the broad applicability of BYND.

In the following, we will use 1000 time steps of RT-TDDFT data of Cd$_{38}$Se$_{38}$-ZnPc-32(NH$_2$CH$_3$) as reference. As we show in Fig. 1a, this signal length is far too short for techniques like Fourier analysis or CS to give any meaningful results. In fact, a standard Fourier transform requires the simulation of 20,000 time steps in order to yield the desired resolution.

Attempts to model the spectral features with the frequencies from SMA in a linear prediction fashion utterly fail due to the number of excitations far exceeding the number of available data points. As a consequence meaningful model parameters (amplitudes) are not extractable even by applying regularization techniques. In other cases the model frequencies from SMA might be in the wrong place and linear prediction alone even with sufficient data points is unable to extract all information. We illustrate these two cases in Fig. 2. In (a), the short-time signal is accurately reproduced, however, the parameters are under-determind and there are many ways to reproduce the signal. It is not possible to select a meaningful spectrum. The bright states are completely absent. In (b), there exists no proper solution at all as there is no SMA frequency available at the position of the first bright feature. All these considerations ultimately lead to the necessity of optimizing the SMA frequencies and to restrict their number.

### Narrow feature selection

Our method is based on the realization that the spectrum of large systems can be separated in a sparse part and continuum part. It is important to realize that the SMA is accurate enough to give us an estimate of how many narrow features should be present, where the narrow features are located (up to 0.5 eV accuracy), how many continuum states are present and in which frequency range the continuum is. Therefore, our decisive step is to use the SMA as an initial guess (Fig. 3a). We select the initial set of narrow features by selecting each frequency for which the SMA transition dipole moment is above a certain threshold. The threshold needs to be chosen according to ensure that only bright excitations are included. In our example we use a threshold of 1.5 a.u. for the intensity (Fig. 3b).

### Narrow feature optimization

The task of finding a set of optimal frequencies $\omega_k$ translates to finding a signal $f^{\text{sparse}}$ which minimizes the error with respect to the short-time dynamics dipole target signal $y$. For this purpose, $f^{\text{sparse}}$ at a certain time step $t_i$, can be defined as

$$f_{\lambda\mu}^{\text{sparse}}(A_k^{\lambda\mu}, \omega_k, t_i) = -\sum_k A_k^{\lambda\mu} \sin(\omega_k t_i),\tag{1}$$

where we make use of the fact that all excitation modes start with an in-phase oscillation right after a sharp $\delta$-pulse[43]. Note, for cases where the electric field is parallel with the dipole operator, we are able to employ a non-negative constraint on the amplitudes.

The first step of our algorithm is to determine the amplitudes $A_k^{\lambda\mu}$ of our target frequencies $\omega_k$. For this purpose we make use of ridge regression[44], also known as Tikhonov regularization[45],

$$\min \frac{1}{n}\sum_i ||y_i^{\lambda\mu} - f_{\lambda\mu}^{\text{sparse}}(A_k^{\lambda\mu}, \omega_k, t_i)||_2^2$$
$$+ \alpha_{\text{sparse}}||f^{\text{sparse}}||_2^2.\tag{2}$$

Here, $\alpha_{\text{sparse}}$ is the regularization coefficient (for discussion on how to choose $\alpha_{\text{sparse}}$, see Supplementary information section 4). Note that, in contrast to methods like CS, we do not need to enforce sparsity here as our SMA initial guess provides us with a good approximation of how many narrow features should be present. In principle, one could use CS or MUSIC for sparse feature extraction, however, we find that direct optimization of SMA provides more accurate results (see Supplementary Figs. 8–12).

---

## BOX 1:

# Line-search

1: initial guess for $\omega_k$ from SMA
2: **While** not converged **do**
3:     $\mathbf{A}_k^{\lambda\mu} \leftarrow \min\frac{1}{\mathbf{n}}\sum_i||\mathbf{y}_i^{\lambda\mu} - \mathbf{f}_{\lambda\mu}^{\text{sparse}}(\mathbf{A}_k^{\lambda\mu},\omega_k,\mathbf{t}_i)||_2^2 + \alpha_{\text{sparse}}||\mathbf{f}^{\text{sparse}}||_2^2$
4:     **if** iteration = 1 **then**
5:         $\Delta\omega \leftarrow \Delta\omega_{\text{init}}$
6:         randomly modify $\mathbf{A}_k^{\lambda\mu}$
7:     **else**
8:         $\Delta\omega \leftarrow \Delta\omega_{\text{def}}$
9:     **for** $\omega_i \in \{\omega_1,...,\omega_k\}$ **do**
10:         **for** $\omega \in \{\omega_i - \Delta\omega,...,\omega_i,...,\omega_i + \Delta\omega\}$ **do**
11:             $\mathbf{f}_{\lambda\mu} \leftarrow -\sum_{k\neq i}\mathbf{A}_k^{\lambda\mu}\sin(\omega_k\mathbf{t}_i) + \mathbf{A}_i^{\lambda\mu}\sin(\omega\mathbf{t}_i)$
12:             Compute $\mathbf{L}(\mathbf{A}_k^{\lambda\mu},\mathbf{A}_i^{\lambda\mu},\omega_k,\omega)$,    $k\neq i$
13:         $\omega_i \leftarrow \min(\mathbf{L})$
14:         $\mathbf{A}_k^{\lambda\mu} \leftarrow \min\frac{1}{\mathbf{n}}\sum_i||\mathbf{y}_i^{\lambda\mu} - \mathbf{f}_{\lambda\mu}^{\text{sparse}}(\mathbf{A}_k^{\lambda\mu},\omega_k,\mathbf{t}_i)||_2^2 + \alpha_{\text{sparse}}||\mathbf{f}^{\text{sparse}}||_2^2$

Finding the optimal frequencies $\omega_k$ is a non-linear optimization problem[46] and can be solved efficiently by performing a line-search around the initial guess for these frequencies. Our algorithm aims to minimize the following objective function,

$$L\left(A_k^{\lambda\mu},\omega_k\right) = \sum_{\lambda\mu}\sum_i ||y_i^{\lambda\mu} - f_{\lambda\mu}^{\text{sparse}}(A_k^{\lambda\mu},\omega_k,t_i)||_2^2 \\ + \beta\sum_i A_k^{\lambda\mu}||\sin(\omega_k t_i) - \sin(\omega_k^{\text{init}} t_i)||_2^2, \quad (3)$$

where the first term measures the error with respect to the target signal. The last term acts as a penalty on frequencies which are too far away from their initial guess $\omega_k^{\text{init}}$ with $\beta$ determining the strength of the penalty. Our procedure is realized as a greedy-algorithm[47], which means our algorithm starts with the frequencies which have the highest amplitude and performs a line-search with a frequency search space $\pm\Delta\omega$ around the initial frequency. If a minimum is found the algorithm updates the old value with the newly found optimum frequency value and performs an additional amplitude adjustment step. It then moves forward to the next frequency. When all frequencies have been updated we start again by finding optimum amplitudes for the new set of frequencies. Both steps, amplitude adjustment and line-search are repeated until frequencies and amplitudes are converged. The entire procedure is described in Box 1. For more details the reader is referred to the discussion in section 1 of our Supplementary information.

As one can see from Fig. 3c, our procedure is able to successfully recover the narrow features in $Cd_{38}Se_{38}$-ZnPc-32($NH_2CH_3$). It should be emphasized that there are, in principle, infinitely many sets of frequencies which minimize the objective function $L$. By starting with a somewhat-accurate initial guess for the number of frequencies, we dramatically reduce the number of possible solutions. Only through this initial guess are we able to locate the correct position of the narrow features within the quasi-continuum of excitations. For the sake of simplicity in this demonstration of our approach, we have set $\beta$ to zero in all our test scenarios. Another possible simplification is to start only with signal components where $\lambda = \mu$. We observe that this can improve convergence behaviour by introducing more constraints on the feature space, only allowing non-negative amplitudes.

## Relaxation of quasi continuum

After optimization of the narrow features, we calculate the residual between the target signal and $f^{\text{sparse}}$,

$$y_{\lambda\mu}^{\text{cont}}(t) = y_{\lambda\mu}(t) - f_{\lambda\mu}^{\text{sparse}}(t). \quad (4)$$

By subtracting $f^{\text{sparse}}$ from our target, $y^{\text{cont}}$ contains only information about the continuum region of the spectrum. We now make use of the fact that the SMA contains also information about the spectral density of the continuum region and perform an additional regression in order to obtain the correct amplitudes for the continuum,

$$\min\frac{1}{n}\sum_i ||y_i^{\text{cont},\lambda\mu} - f_{\lambda\mu}^{\text{cont}}(A_k^{\lambda\mu},\omega_k,t_i)||_2^2 \\ + \alpha_{\text{cont}}||f^{\text{cont}}||_2^2. \quad (5)$$

Note that, in this linear prediction, the index $k$ indicates the frequencies obtained from the SMA. As the target signal does not contain any narrow feature components, we set the regularization coefficient $\alpha_{\text{cont}}$ to the default value of 100. Our final reconstructed dipole signal is then given by

$$f_{\lambda\mu}(t) = f_{\lambda\mu}^{\text{cont}}(t) + f_{\lambda\mu}^{\text{sparse}}(t). \quad (6)$$

As we show in Fig. 3d, our algorithm is able to accurately reproduce the exact short-time dynamics signal. We obtain amplitudes and frequencies of the bright states as well as correct amplitudes for the continuum region. We would like to highlight that our algorithm is completely independent of the underlying electronic structure code and can be realized in a Python implementation which easily runs on standard local desktop and laptop computers.

## Convergence and accuracy

Figure 4 shows the convergence of the calculated absorption spectrum with respect to the number of time steps of the target electronic dipole signals for three different systems $Cd_{38}Se_{38}$-ZnPc-32($NH_2CH_3$), $Cd_{38}Se_{38}$-ZnPc-DPA-32($NH_2CH_3$) and $Cd_{33}Se_{33}$/$Zn_{93}S_{93}$-2(ZnPc). These systems demonstrate how our method performs with different types of spectra and signals. For $Cd_{38}Se_{38}$-ZnPc-DPA-32($NH_2CH_3$) we expect the emergence of additional narrow features due to the presence of the DPA molecule on top of ZnPc (Fig. 4). On the contrary, the $Cd_{33}Se_{33}$/$Zn_{93}S_{93}$-2(ZnPc) system has two ZnPc molecules and a significantly larger nanocrystal size. This larger nanocrystal leads to more blurring of the bright, localized excitations into the continuum. In addition, the two ZnPc molecules mimic a higher surface coverage and are on top bound to two different facets of the nanocrystal. Overall this system consists of 7572 electrons and is thus roughly two times larger than the other two nanocrystals and thus can be regarded as a highly challenging test case for our method.

To further support our visual analysis, we calculate the Pearson correlation coefficient ($\rho$) between various approximate methods and the 20,000 time step reference spectrum in the range of 1 to 12 eV. This allows us to quantify similarities between two spectra regarding overall shape and intensity across a broad frequency range. For this purpose, we utilize our two largest nanocrystal systems, $Cd_{33}Se_{33}$/$Zn_{93}S_{93}$-2(ZnPc) and $Cd_{33}Se_{33}$/$Zn_{93}S_{93}$-2(ZnPc)-DPA (see Supplementary Fig. 20), and average over the obtained correlation coefficients. Due to their high spectral density and spectral narrow feature characteristics, these systems pose significant challenges for any super-resolution technique. Figure 5 shows the results for BYND compared to other super-resolution approaches.

While the Pearson correlation is useful to quantify spectral similarity, we would like to emphasize that in many cases visual inspection reveals that BYND spectra are more similar to the converged spectra

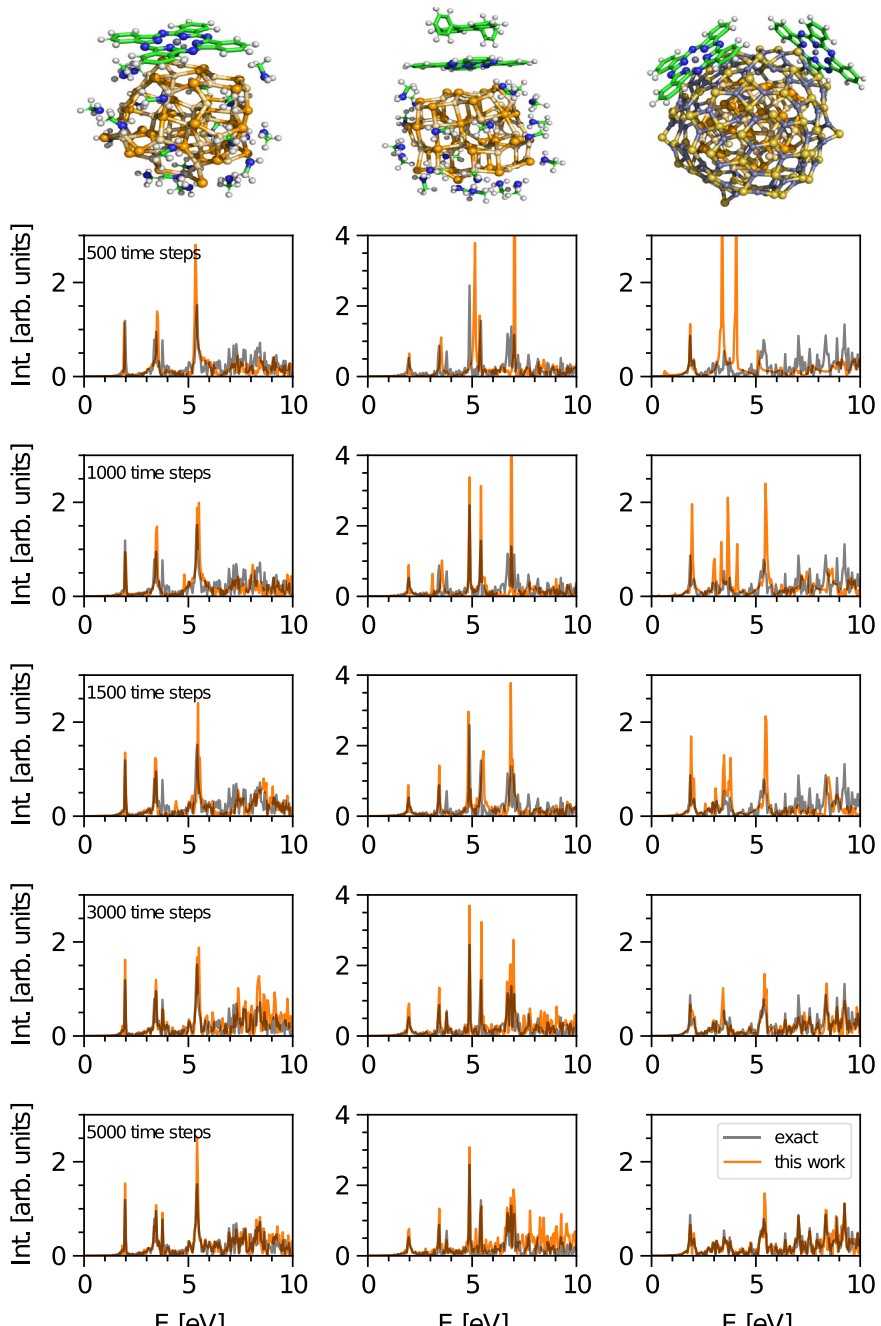

**Fig. 4 | Convergence behaviour with respect to the number of data points.** We use Cd$_{38}$Se$_{38}$-ZnPc-32(NH$_2$CH$_3$) (left), Cd$_{38}$Se$_{38}$-ZnPc-DPA-32(NH$_2$CH$_3$) (middle) and Cd$_{33}$Se$_{33}$/Zn$_{93}$S$_{93}$-2(ZnPc) (right) as a prototypical examples. We show the result for the absorption spectrum by varying the length of the short-time dynamics dipole signals between 500 and 5000 time steps. The reference RT-TDDFT absorption spectrum was simulated with in total 20,000 time steps. For details regarding the input frequencies, the reader is referred to Supplementary Tables 2–5. Intensity was abbreviated with Int.

than the Pearson coefficient suggests (see methods section for more details). Thus, Fig. 5 serves as something of an upper bound on the error of BYND compared to other methods.

We have additionally demonstrated the accuracy of BYND for purely molecular systems (see Supplementary Fig. 15).

**Observations, trends and limitations**
As one can see in Fig. 4, we begin to obtain relatively accurate spectra compared to RT-TDDFT for our simplest system starting at only 500 time steps. Generally, all narrow features are reproduced for all test systems. Unsurprisingly, more challenging systems, namely Cd$_{33}$Se$_{33}$/Zn$_{93}$S$_{93}$-2(ZnPc), require more data points to achieve high accuracy results. However, we note that even for this system the spectrum is well reproduced using only 1500 time steps.

In cases where the splitting between the bright features is small, BYND requires more short-time data to fully resolve these details. For example, for the bright states at an excitation energy of around 3.5 eV in the Cd$_{38}$Se$_{38}$-ZnPc-32(NH$_2$CH$_3$) system, our algorithm predicts one single highly bright feature instead of the two exact, less bright excitations. With such a small number of data points our algorithm is not able to distinguish between these two frequencies and more time steps are needed to resolve them. We first observe the emergence of the second bright excitation upon including 3000 time steps, which is still roughly seven times shorter than the signal required for standard

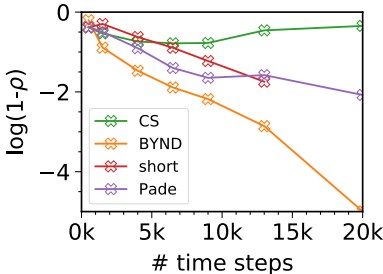

**Fig. 5 | Averaged Pearson correlation coefficient of Cd$_{33}$Se$_{33}$/Zn$_{93}$S$_{93}$-2(ZnPc) and Cd$_{33}$Se$_{33}$/Zn$_{93}$S$_{93}$-2(ZnPc)-DPA.** The Pearson correlation coefficient ($\rho$) is displayed as $\log(1-\rho)$. The reference spectrum is the Fourier transform from a 20,000 time step RT-TDDFT simulation. We compare the performance of BYND with compressed sensing (CS), Fourier-Padé (Pade), and Fourier transformation of a short-time signal (short). The correlation coefficients have been calculated for a spectral window from 1 to 12 eV. For technical reasons ($\log(1-\rho) \to -\infty$), we have omitted the final point of the Fourier transform of the short-time signal.

Fourier analysis. This is a general trend, and we obtain detailed resolved narrow features for all test systems when using 3000 time steps.

Similar observations can be made regarding the relative intensities. The narrow features can be clearly distinguished from the quasi-continuum background for all lengths of the short-time signal, but finding the correct relative intensities requires more time steps. The correct relative amplitudes of the narrow features are reproduced using 3000 time steps for all considered systems. The improvement coincides with a significant better assignment of continuum amplitudes. Continuum amplitudes are obtained from the residual signal $y_{\lambda\mu}^{\mathrm{cont}}$ which is described in eq. (4). As $f_{\lambda\mu}^{\mathrm{sparse}}$ becomes more and more accurate $y_{\lambda\mu}^{\mathrm{cont}}$ will be as well. On the other hand, an overestimation of amplitudes for bright excitations thus naturally leads to underestimation of the continuum region.

Going from 500 to 5000 time steps shows a clear convergence behaviour in accuracy. At 5000 time steps, we achieve already almost excellent agreement with the exact result which is still four times less data points compared to the full RT-TDDFT run. However, while significantly mitigated, errors in amplitude are still evident. The fact that BYND shows a clear convergence is supported by Fig. 5 where we compare the similarity with the long-time reference. Convergence towards the exact results underlines that BYND is not an approximate method. Provided with enough data points, BYND will yield the exact time dynamics. Thus, it demonstrates that BYND also fulfills the Thomas-Reiche-Kuhn sum rule[48–50] for the oscillator strength. Fourier-Padé approximation and CS do not show a systematic convergence behaviour. Both typically work best for a few well-separated narrow features which does not hold true anymore for the densely populate spectra of our systems[18,51]. Contrary, BYND shows a significant better correlation with the reference spectrum for any number of time steps from 1500 onwards.

BYND is capable of describing the excited state spectra not only of nanocrystals but also for a broad variety of other systems. The field of dye-sensitized solar cells[52], solar batteries[53], energy transfer[54] or catalysis[55], as well as chemical sensing[56] are just a few examples where BYND can be applied. To highlight this aspect, Fig. 6 displays spectra of a molecular aggregate, a nanotube, and two surface systems which are all accurately reproduced. Even when broad and narrow features coincide, as evident in Fig. 6a, d, BYND yields reliable results. Furthermore, the broad features in Fig. 6b/c at around 7 eV emerges from the continuum amplitude fitting and was not part of the narrow feature optimization. We conclude that if the dominant narrow features are correctly reproduced, additional broad features can be captured by the continuum fitting procedure. Our observations indicate that,

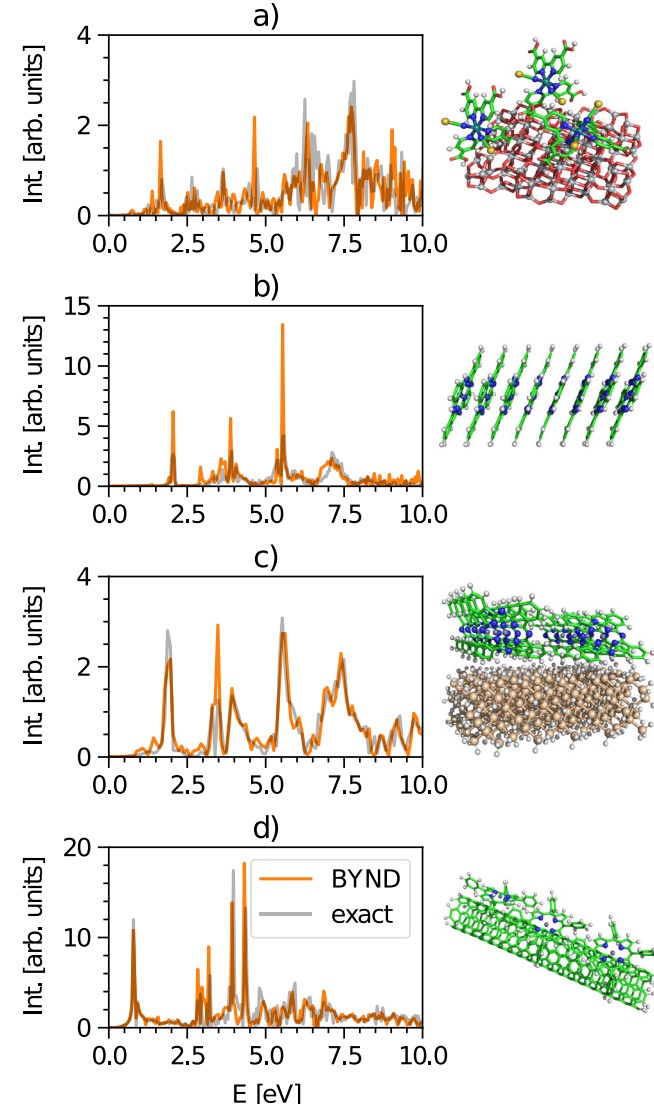

**Fig. 6 | Additional systems. a** Cis-[Ru(4,4'-COOH-2,2'-bpy)$_2$(NCS)$_2$] on an anatase (101) cluster as an example for a dye-sensitized solar cell[65,66]. **b** Molecular ZnPc j-aggregate. **c** ZnPc film on a Si (111) surface. **d** Zinc-porphyrin molecules on a carbon nanotube[54]. Time steps: **a** 2500, **b** 1500, **c** 2000 and **d** 3000. Time steps have been chosen to provide a good trade-off between accuracy and minimizing the amount of data. Due to the large system size, we only show system (**c**) with a reference spectrum of 10,000 time steps; otherwise, we use 20,000 time steps. Intensity is abbreviated with Int.

separating the signal into sparse and continuum components is robust enough even for challenging spectral patterns. Thus this approach should not be purely restricted to electronic structure applications only. As long as the signal is separable into continuum and sparse parts by any kind of initial guess, BYND should be able to yield the correct spectral information.

In the case of electronic excitation spectra, BYND's performance will generally depend on the quality of the SMA input frequencies used to identify the sparse contribution. In addition to errors in these frequencies themselves, there is also the possibility that the SMA generates too few or too many frequencies. We demonstrate in Supplementary Figs. 2 and 3 that the case where too many narrow feature frequencies are selected is usually not of concern. When too few frequencies are selected, BYND can possibly miss narrow features. The same holds true if $\omega_{\mathrm{init}}$ is too small and input frequencies are too far away from their target. We find that measuring the quality of the fit

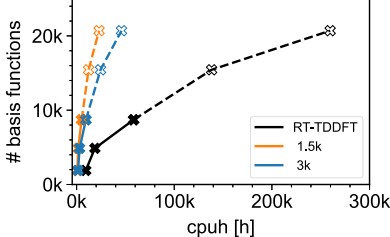

**Fig. 7 | Possible system sizes.** We show the required time for BYND for different time steps 1500 and 3000 respectively. Calculations have been performed on Intel(R) Xeon(R) Silver 4210R CPUs @ 2.40GHz. The total time of our approach consists of the time required for the SMA plus the time for the three time-dependent density functional theory (RT-TDDFT) short-time dynamics runs. For the high resolution long-time dynamics simulation, the total required computational time is the cost for three RT-TDDFT runs with 20,000 time steps each. Unfilled markers and dashed line represent additional nanocrystal systems (see Supplementary Fig. 21) where we have used the RT-TDDFT wall-time estimates provided by FHIaims to predict their computational cost.

between the target and BYND signals provides a useful tool for identifying both cases (see discussion in Supplementary information section 2). Another limitation of BYND arises from the TDDFT dipole signal's lack of information about "dark" excitations; thus, BYND in the context of electronic excitations is limited to excitation energies with non-vanishing oscillator strengths.

Overall, our analysis shows that BYND is able to correctly predict the full excitation spectrum of large systems. Narrow features embedded in the continuum are clearly evident; bright molecular features, CT features, and contributions from the continuum are all well reproduced. This is achieved while significantly reducing the required computational workload. The range of 500 to 1500 time steps corresponds to a speed up by a factor of 13 to 40 compared to high resolution results. For reference, this cost reduction brings the calculation of a full TDDFT spectrum for a large system close to the computational cost of a standard ground state geometry optimization. Excellent agreement is then achieved by including more data points.

**System size considerations**
In order to give a perspective on which system sizes are possible with BYND, we display in Fig. 7 the scaling of BYND's computational cost with system size. The filled data points represent full simulations of f-cororene, $Cd_{38}Se_{38}$-ZnPc-32($NH_2CH_3$), and $Cd_{33}Se_{33}/Zn_{93}S_{93}$-2(ZnPc). Unfilled markers represent additional nanocrystal systems (see Supplementary Fig. 21) where we have used the RT-TDDFT wall-time estimates provided by FHIaims to predict their computational cost. At a given walltime, BYND enables the simulation of significantly larger systems than a standard RT-TDDFT run, even when using an input signal of 3000 time steps. To quantify this further, for $Cd_{33}Se_{33}/Zn_{93}S_{93}$-2(ZnPc) – one of our largest nanocrystals – we achieve very good narrow feature accuracy with a signal length of only 1500 time steps. For this number of time steps, BYND needs only 5138 CPUh, compared to the 58,359 CPUh required for the full long-time dynamics run – a reduction in required computational time by factor of 11.3.

In this work we showed how approximate frequency space results can be combined with short-time dynamics simulations in order to accurately capture narrow features and a quasi-continuum of states for large systems. Due to the ability of BYND to use only short-time dynamics, we are able to significantly reduce the computational time which is needed for the underlying electronic structure simulations. For one of our highly challenging systems we observe a reduction by a factor of 11. The reduction of computational time is due to two key components of our approach. First, we use the SMA as an estimate for

how many narrow features can be expected and which frequencies they have. Second, we use this information to further optimize their position and amplitudes by minimizing the error with respect to the short-time dynamics signal which on its own would have insignificant resolution to capture the spectrum. Thus, our approach allows researchers to understand the electronic properties of large systems which were previously computationally inaccessible. In contrast to methods such as filter diagonalization[57], which only shows promising results if the spectrum is not too dense[51], BYND is explicitly designed to work with high spectral densities. Further, we would like to emphasize that BYND is not an approximation: if enough data is provided, the results will always converge towards the full-time dynamics of the chosen electronic structure method. This is in contrast to other methods such as simplified TDDFT[58], simplified GW/sBSE[59], or TD-INDO/S[60] which employ approximations to the electron interaction integrals in order to achieve computational speedup. To increase the data available without computing more time steps, future work on BYND is aimed towards including quadrupole or higher multipole moments. More data points will then enable the use of even shorter time dynamics. Improvement in accuracy could be achieved by using the Casida equations to explicitly describe just the first few excited states, which can be then fixed in our non-linear optimization to yield an even more efficient localization of the remaining narrow features. Furthermore, we see potential in improving our line-search routine by employing advanced machine-learning techniques, which may allow us to further increase the range of the spectrum covered by the sparse signal. Further one can use SMA results from semi-local exchange-correlation kernels to approximate hybrid TDDFT or GW results, thus saving additional simulation time.

In conclusion, we have combined frequency domain results with exact short-time dynamics in order to create a super-resolution technique (BYND) which allows for the ab initio description of the entire excitation spectrum for systems which are beyond the system size boundaries of current electronic structure methods.

## Methods
**TDDFT simulations.** The time-dependent Kohn-Sham states are explicitly propagated in time under the influence of an electric field, $E_\lambda(t) = V_\lambda\delta(t)$[34,35]. Once the time-dependent dipole moment ($\mu_\nu(t)$) is obtained from the simulation, one can use the polarizability tensor in frequency space,[12,61]

$$\alpha_{\lambda\nu}(\omega) = \frac{1}{V_\lambda}\int_0^\infty dt e^{-i\omega t}\left[\mu_\nu(t) - \mu_\nu(t_0)\right] \quad, \tag{7}$$

to calculate the final excitation spectrum[61]

$$S(\omega) = \frac{2\omega}{3\pi}\mathrm{Tr}\{\Im[\alpha(\omega)]\} \quad. \tag{8}$$

All TDDFT calculations for our nanocrystal systems have been carried out using the FHIaims[62] program. Exchange-correlation interactions have been treated using the PBE[63] functional. *Light tier1* settings have been used for the integration grid and basis set. All RT-TDDFT[61] calculations have been performed with a time step of 0.2 a.u. and an electric field strength of 0.01 a.u. Total simulation time was 4000 a.u.

**Quantifying similarities between two spectra.** In order to quantify the similarity between two spectra, we make use of the Pearson correlation coefficient,

$$\rho = \frac{\sum_i (x_i - \bar{x})(y_i - \bar{y})}{\sqrt{\sum_i (x_i - \bar{x})^2}\sqrt{\sum_i (y_i - \bar{y})^2}} \quad. \tag{9}$$

Here, both spectra ($x$ and $y$) are represented as vectors with their corresponding mean values $\bar{x}$ and $\bar{y}$. We can interpret Eq. (9) as the overlap of the variation of the spectrum from its average; thus, the overlap of a completely smooth distribution will be exactly zero. We find that the Pearson correlation is therefore more sensitive when comparing spectra obtained from very short-time signals, as opposed to, for example, the spectral angle mapper which only accounts for the overlap. One limitation of quantifying spectral overlaps is that spectral shifts and differences in peak positions may not adequately captured. Eq. (9) compares the two spectra bin by bin, which means that if two very narrow features are just shifted slightly, their overlap would be zero. However, a visual inspection would clearly indicate a large similarity in this situation. In order to mitigate this effect, we convolute the obtained spectrum with a Gaussian function, which effectively spreads out the spectral peaks and increases their width. Note that this Gaussian broadening is not applied to our 20,000 RT-TDDFT reference. For each method (see Fig. 5) we apply a range of broadening factors for each number of time steps and calculate the correlation coefficient between the broadened spectrum and the reference spectrum. We then pick the broadening that gives the optimal Pearson coefficient.

In general, numerous peaks as well as broad features make quantifying the difference between very complex spectra quite challenging. As a consequence, quantifying the difference between two spectra as a whole may lack sensitivity. Nevertheless, this analysis is an excellent support for Fig. 4 where we also provide visual representations of the obtained spectra to allow for more comprehensive understanding.

**SMA.** In this approximation, the electronic excitation energies in frequency space are simply given by

$$\omega_i = \sqrt{(\epsilon_a - \epsilon_i)^2 + 4(\epsilon_a - \epsilon_i)\langle ia|f_{Hxc}|ia\rangle} \quad , \tag{10}$$

with $\epsilon_a$ and $\epsilon_i$ being the eigenstate energies of the $a$-th virtual and $i$-th occupied Kohn-Sham state while $f_{Hxc}$ denotes the Hartree and exchange-correlation kernel. Thus, the SMA gives us a simple analytical expression for obtaining a large number of excitation energies without the necessity of solving the Casida equations directly. Strictly speaking, the SMA is only exact if the single-particle excitations show vanishing overlap, and thus in realistic systems the SMA is error prone and can only serve as a first approximate step. However, it remains a significant improvement over, for example, just using the ground state spectrum as it contains a great deal of information about the relative location of the bright states.

### Reporting summary
Further information on research design is available in the Nature Portfolio Reporting Summary linked to this article.

## Data availability
SMA initial guess frequenies, xyz-structure files and TDDFT input files are provided in the Supplementary Information/Supplementary Data 1 file. Source data for Figs. 1–7 are provided with this paper in form of a Source Data file. Source data are provided with this paper.

## Code availability
The code and a tutorial on how to use it can be obtained from our GitHub repository (BYND)[64].

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

## Acknowledgements

M.K. acknowledges support from the German Research Foundation (DFG, KI 2558/1-1, 505191319). Further, M.K. would like to thank Prof.

Harald Oberhofer and Cristina Grosu for their valuable input and support. T.V.V. acknowledges support from the National Science Foundation (Award no. CHE-2154938). This work used expanse at the San Diego Supercomputing Center through allocation CHE200006 from the Advanced Cyberinfrastructure Coordination Ecosystem: Services & Support (ACCESS) program, which is supported by the National Science Foundation grants (Award no. OAC-2138259, OAC-2138286, OAC-2138307, OAC-2137603, and OAC-2138296).

## Author contributions

M.K. performed all electronic structure calculations, designed the algorithm and performed all necessary code implementations. M.K. also wrote the manuscript. E.A. helped editing the manuscript and contributed fruitfully in various discussions. A.B. carried out test calculations on small molecular systems. T.V.V. edited the manuscript and helped with algorithm design.

## Competing interests

The authors declare no competing interests.
