## [Peer Review File · Nature Communications]

Super-resolution techniques to simulate electronic spectra of large molecular systemsReviewer #1 (Remarks to the Author):

In this manuscript, the authors present a post-processing algorithm that can considerably reduce the computational cost of real-time time-dependent DFT and other similar methods that are used to obtain excited-state properties of a wide range of systems. Although deceptively simple, this algorithm seems to succeed where other proposed methods have failed. Coupled with the superior scaling with system-size of TD-DFT, it greatly extends the size of the systems that can be successfully tackled by existing codes.

I found the manuscript well written, providing a balanced level of detail. I thus recommend this manuscript for publication in Nature Communications after the following comments are addressed.

1) The dipole strength function $S(\omega)$ defined in eq.3 is known to obey the Thomas–Reiche–Kuhn f-sum rule for the number of electrons:

$$N = \int d\omega S(\omega)$$

where N is the number of electrons in the system. This is a simple condition that could be used to check the quality of the linear prediction. Better yet, maybe it would be possible to modify the algorithm to impose that the sum rule is fulfilled. This could potentially improve the predicted peak intensities.

2) Line 8 of the line-search algorithm pseudo-code states that there's a loop over frequencies belonging to the $\{-\Delta\omega, \dots, \omega_i, +\Delta\omega\}$ interval. It seems to me that this is inconsistent with what is written elsewhere in the manuscript. Instead, I would expect the frequencies to belong to the $\{\omega_i - \Delta\omega, \dots, \omega_i, \dots, \omega_i + \Delta\omega\}$ interval. Probably this is what the authors actually meant, but then the notation is confusing. In the same pseudo-code, it is also unclear what is the value of $\Delta\omega$ for iterations after the first one.

3) I find the section on timing and speed-up somehow unnecessarily long. Since the cost of a time-dependent calculation, excluding initialization and finalization, usually scales linearly with the number of time-steps, it seems quite straightforward to estimate the savings in computational cost. It is also not clear to me what new information is provided by showing the estimated savings for larger systems. Although not mandatory, I would suggest the authors to shorten this section.

4) Finally, both in the manuscript and supporting information, the chosen line-style and line-width of the figures makes it often difficult to compare two curves. For example, in Fig. 4, it's difficult to judge on the agreement of peak intensities for some of the excitations. Also in Fig. 4, the inconsistency of the y-range in the middle-column plots is confusing and makes the convergence of the spectra with respect to the number of time-steps harder to gauge.

Reviewer #1 (Remarks on code availability):

The code structure and style could benefit from some improvements, but it does provide a reasonable amount of documentation, including README files with some usage instructions.

No installation instructions are provided. Data used in the paper is provide along the code, which allows to check that the paper results are reproducible.

Reviewer #2 (Remarks to the Author):

This manuscript describes techniques to compute electronic spectra of complex molecules from real-time time-dependent density functional theory by circumventing the spectral resolution limitations imposed by finite propagation times. That in itself is highly important and timely, because of a recent increase in the application of real-time time-dependent DFT owing to its excellent balance between accuracy and computational affordability, especially when scaling to large numbers of CPUs or GPUs. The manuscript describes the techniques and their novelty, assesses numerical convergence and parameter choices, and studies timing and speedups. The work provides exciting insight into these aspects by studying a "challenging set of large systems" that mix bulk, surface, and molecular states and comprise of thousands of electrons/electronic states. For all this I highly commend the authors!

However, I must admit that I am split about this manuscript. On the one hand, the theoretical and methodological advance is clear. Potential applications of this technique are likely plenty, given the increased interest in real-time TDDFT nowadays. (In fact, if anything the thorough description of the method in this manuscript could probably only be improved by more clearly discussing the limitations of the approach (bulk materials? defects? ...) because the advantages are clear and well discussed.) For these reasons I am not concerned about the impact.

My concern is that the direct impact of this work is likely restricted to the community of first-principles electronic-structure theory. Experimentalists will likely not take much from this work, since they will not implement or use this technique themselves. Furthermore, also the large molecular systems studied in this work seem to exclusively serve the purpose of outlining the advantages of the method, but the results do not seem to show any unexpected science about these molecules themselves? I might, of course, be wrong with this conclusion and in that case encourage the authors to clarify that scientific insight in a revision. In addition, choosing prototypical "large systems" is absolutely fine and sufficient to convey the usefulness of the method developed here, however, it does restrict the interest of this manuscript more to the theoretical first-principles electronic structure theory community in my opinion.

For these reasons I recommend publication of this manuscript, however, a more specialized method focused journal may be more appropriate for this work in its present form.

Reviewer #3 (Remarks to the Author):

This manuscript describes a method for improving the efficiency of calculating spectra for large systems, based on TDDFT. The method seems effective, although I have some comments and concerns. Additionally, the manuscript does not seem like a typical Nature Communications paper. My specific comments are:

- This work is more methodological and detailed than most Nature Communications papers, and it may not have the broad interest required for this journal. However, I did not initially realize that the code is shared (perhaps this would be obvious in the published version but I did not see it in the manuscript). While this does potentially help the broad interest, it is still a detailed, methodological paper.
- The authors compare computational effort between their method and an “exact” results from 20,000 timesteps to obtain a speedup. However, given that their method also somewhat lowers accuracy, a more fair comparison would be to check how many timesteps it takes to get the same level of accuracy from existing methods. A plot of accuracy vs. CPU time for traditional methods as compared to the new method would be useful. This should be fairly easy to create from the data the authors already have.
- In Figure 2b, the authors suggest that linear prediction fails when there is a frequency outside of the frequencies predicted by SMA. I do not see concrete evidence that their method solves this particular issue.
- Does the alpha value affect the results? Is there any scaling of the raw signal? How will future users choose an appropriate alpha?
- "Provided with enough data points, BYND will yield the exact time dynamics." This seems sensible, but would also be easy for the authors to check to prove it.
- “Our method is based on the realization that the spectrum of large systems can be separated in a sparse part and continuum part. “ This separation may not always be clear. For example, if a system has several broad features, will the current method be effective?
- It is not always easy to see both the black and orange lines, particularly when the orange peaks are larger than the black peaks. Would some adjustment of line thickness/style, translucency, etc. make this easier to see?
- The writing is generally fairly good. However, there are a fair number of typos (e.g., duo instead of due). Also, there is a sentence fragment: “Thus, adding only a vanishing amount of computational time.”

Reviewer #3 (Remarks on code availability):

From looking through the repository, it appears to be reasonably user-friendly.

Reviewer #4 (Remarks to the Author):

This manuscript describes a new practical algorithm for computing the electronic absorption spectrum of large systems with many transitions via TDDFT. In short, the method exactly computes a subset of the strongest transitions, and then performs a fitting procedure to approximate the rest. The results are very encouraging. The paper is well written, detailed, and interesting. This work is certain to be of interest to other researchers interested in computing electronic spectra. I recommend publication after the authors have revised the manuscript to address the following questions.

- 1) Is there any way to know how accurate the spectrum is without comparing to the exact spectrum? That is, could a user estimate the error in their spectrum without computing a better one?

2) Similarly, it would be interesting to quantify the convergence of the spectra in figure 4 with respect to the propagation time. Does the error decrease predictably with increasing simulation time, and if so, to what order does the error scale?

3) Will SMA necessarily provide a good prediction of the number of narrow features? For example, plasmonic excitations are collective motions of many electrons, and therefore typically exist as superpositions of many amplitudes. Would the approach describe in this work be able to handle excitations like this?

Answers to reviewers: Broad Yet Narrow: Super-resolution techniques to simulate electronic spectra of large molecular systems

Matthias Kick^{*†}, Ezra Alexander[†], Anton Beiersdorfer[‡], and Troy Van Voorhis[†]

[†]Department of Chemistry, Massachusetts Institute of Technology, Cambridge, Massachusetts 02139, USA

[‡]Technical University of Munich, Lichtenbergstrasse 4, Garching 85747, Germany

June 14, 2024

Reviewer #1 (Remarks to the Author):

In this manuscript, the authors present a post-processing algorithm that can considerably reduce the computational cost of real-time time-dependent DFT and other similar methods that are used to obtain excited-state properties of a wide range of systems. Although deceptively simple, this algorithm seems to succeed where other proposed methods have failed. Coupled with the superior scaling with system-size of TD-DFT, it greatly extends the size of the systems that can be successfully tackled by existing codes.

I found the manuscript well written, providing a balanced level of detail. I thus recommend this manuscript for publication in Nature Communications after the following comments are addressed.

1) The dipole strength function $S(\omega)$ defined in eq.3 is known to obey the Thomas-Reiche-Kuhn f-sum rule for the number of electrons:

$$N = \int d\omega S(\omega)$$

where N is the number of electrons in the system. This is a simple condition that could be used to check the quality of the linear prediction. Better yet, maybe it would be possible to modify the algorithm to impose that the sum rule is fulfilled. This could potentially improve the predicted peak intensities.

This observation is spot-on, and it is indeed the most important constraint that our algorithm is already designed to fulfill. However, we enforce the Thomas-Reiche-Kuhn f-sum rule in the time domain rather than in frequency space. In general, we find that constraining the f-sum rule in the time domain is more accurate for small number of time steps, as we observe the waveform directly.

In the following, we will demonstrate that constraining the f-sum rule in the time domain is equivalent to applying the constraint in frequency space. Due to the linearity property of the Fourier-transform we have a one to one correspondence between amplitudes in time and frequency domain,

$$\sum_i a_i x_i(t) \longleftrightarrow \sum_i a_i X_i(\omega) \quad . \quad (1)$$

By applying a very short electric field pulse we can write the time-dependent dipole moment as a sum of sine waves, $x_i = \sin(\omega_i t)$. Thus, each integral over frequency space will reduce to a sum of amplitudes,

$$\int d\omega \sum_i a_i X_i(\omega) = \sum_i a_i \underbrace{\int d\omega X_i(\omega)}_{=1} = \sum_i a_i = N \quad . \quad (2)$$

*mkick@mit.edu

Thus, by minimizing our loss function,

$$L(A_k^{\lambda\mu}, \omega_k) = \int dt \sum_{\lambda\mu} \|y^{\lambda\mu}(t) - f_{\lambda\mu}^{\text{sparse}}(A_k^{\lambda\mu}, \omega_k, t_i)\|_2^2 \quad (3)$$

$$+ \beta \sum_i A_k^{\lambda\mu} \|\sin(\omega_k t_i) - \sin(\omega_k^{\text{init}} t)\|_2^2, \quad (4)$$

we essentially ensure that our approach fulfills the f-sum rule. This constraint is enforced multiple times during the optimization process, particularly after our algorithm identifies a new frequency minimum (line 14 in Algorithm 1 of the main manuscript).

However, our method is based on that the spectrum of large systems can be effectively separated into a sparse part and a continuum part. Consequently, only the narrow features are included in the frequency optimization algorithm, and thus the sum rule is only imposed on them. This can lead to the described inaccuracy in intensities for a very small number of time steps. Including all frequencies in this step would allow us to correctly predict the amplitudes even for small time steps. However, doing so typically results in a large number of frequencies, which exceeds the number of data points by a significant margin, leading to an under-determined system that makes it nearly impossible to extract meaningful spectra. Thus, narrow feature selection is a necessary preconditioning step that must be taken in order to solve the problem. This trade-off between amplitude accuracy and frequency position is inherent in our approach.

Nevertheless, we have now included a more detailed analysis regarding spectral convergence towards the 20,000 time step reference. This analysis clearly demonstrates that as more periods are sampled, the amplitudes are reproduced accurately, indicating that the f-sum rule constraint is already enforced (see Figure 5 revised manuscript).

We would like thank the reviewer for bringing this to our attention. These considerations have prompted us to adopt a new approach for evaluating the quality of our solution in relation to the number of selected narrow features. This can effectively be achieved by comparing the coefficient of determination (R^2) for various signal lengths. We delve into this topic in more detail in our supplementary information (section 2). We also elaborate more on this aspect in the answer to reviewer four.

2) Line 8 of the line-search algorithm pseudo-code states that there's a loop over frequencies belonging to the $-\Delta\omega, \dots, \omega_i, +\Delta\omega$ interval. It seems to me that this is inconsistent with what is written elsewhere in the manuscript. Instead, I would expect the frequencies to belong to the $\omega_i - \Delta\omega, \dots, \omega_i, \dots, \omega_i + \Delta\omega$ interval. Probably this is what the authors actually meant, but then the notation is confusing. In the same pseudo-code, it is also unclear what is the value of $\Delta\omega$ for iterations after the first one.

This indeed was an inaccuracy in the manuscript. We have made the change from $-\Delta\omega, \dots, \omega_i, +\Delta\omega$ to $\omega_i - \Delta\omega, \dots, \omega_i, \dots, \omega_i + \Delta\omega$, and we have also added a line specifying the value of $\Delta\omega$ for later iterations ($\Delta\omega_{\text{def}}$). Furthermore, we have provided a more detailed explanation in our supplementary information (section 1).

3) I find the section on timing and speed-up somehow unnecessarily long. Since the cost of a time-dependent calculation, excluding initialization and finalization, usually scales linearly with the number of time-steps, it seems quite straightforward to estimate the savings in computational cost. It is also not clear to me what new information is provided by showing the estimated savings for larger systems. Although not mandatory, I would suggest the authors to shorten this section.

We agree that the section discussing timing and speed-up was a bit lengthy. We have shortened this section and now focus solely on aspects related to system sizes.

4) Finally, both in the manuscript and supporting information, the chosen line-style and line-width of the figures makes it often difficult to compare two curves. For example, in Fig. 4, it's difficult to judge on the agreement of peak intensities for some of the excitation. Also in Fig. 4, the inconsistency of the y-range in the middle-column plots is confusing and makes the convergence of the spectra with respect to the number of time-steps harder to gauge.

We have adjusted the transparency values for all figures in which we explicitly compare our approach with the long-time RT-TDDFT simulation. Specifically, in the main manuscript, this adjustment applies to Figure 4. In the supplementary information, it applies to Figures 4, 15, 16, 17, 18, 19, and 20. Additionally, we have modified the y-range in Figure 4 of our main manuscript. Also all new added Figures have been adjusted accordingly.

Reviewer #1 (Remarks on code availability):

The code structure and style could benefit from some improvements, but it does provide a reasonable amount of documentation, including README files with some usage instructions. No installation instructions are provided. Data used in the paper is provide along the code, which allows to check that the paper results are reproducible.

We have added installation instructions for each Python package required to run the code. We provide a brief description of how to install the packages via pip and via the package manager.

In case the user needs more information, we also refer them to the main page of scikit-learn for additional details regarding the installation of this package and its dependencies.

We also agree that the best way to distribute the code would be to have it as its own Python package. We plan to enable this possibility as soon as we add functionality for time-dependent quadrupole signals, for which we are planning a separate publication.

Reviewer #2 (Remarks to the Author):

This manuscript describes techniques to compute electronic spectra of complex molecules from real-time time-dependent density functional theory by circumventing the spectral resolution limitations imposed by finite propagation times. That in itself is highly important and timely, because of a recent increase in the application of real-time time-dependent DFT owing to its excellent balance between accuracy and computational affordability, especially when scaling to large numbers of CPUs or GPUs. The manuscript describes the techniques and their novelty, assesses numerical convergence and parameter choices, and studies timing and speedups. The work provides exciting insight into these aspects by studying a "challenging set of large systems" that mix bulk, surface, and molecular states and comprise of thousands of electrons/electronic states. For all this I highly commend the authors!

However, I must admit that I am split about this manuscript. On the one hand, the theoretical and methodological advance is clear. Potential applications of this technique are likely plenty, given the increased interest in real-time TDDFT nowadays. (In fact, if anything the thorough description of the method in this manuscript could probably only be improved by more clearly discussing the limitations of the approach (bulk materials? defects? ...) because the advantages are clear and well discussed.) For these reasons I am not concerned about the impact.

My concern is that the direct impact of this work is likely restricted to the community of first-principles electronic-structure theory. Experimentalists will likely not take much from this work, since they will not implement or use this technique themselves. Furthermore, also the large molecular systems studied in this work seem to exclusively serve the purpose of outlining the advantages of the method, but the results do not seem to show any unexpected science about these molecules themselves? I might, of course, be wrong with this conclusion and in that case encourage the authors to clarify that scientific insight in a revision. In addition, choosing prototypical "large systems" is absolutely fine and sufficient to convey the usefulness of the method developed here, however, it does restrict the interest of this manuscript more to the theoretical first-principles electronic structure theory community in my opinion.

For these reasons I recommend publication of this manuscript, however, a more specialized method focused journal may be more appropriate for this work in its present form.

While it may initially seem that the impact of this work is confined to the first-principles electronic-structure community, the broader impact stems from the method itself and its core idea of separating the signal into sparse and continuum contributions. BYND can be used in cases where other established super-resolution techniques fail, such as when one wants to obtain sharp features in frequency space while the overall spectrum also contains

broad or “noisy” features. Standard super-resolution techniques such as Compressed Sensing, MUSIC, or Fourier-Padé approximation usually perform best for a few well-separated peaks with low noise levels. Thus, they become unreliable for signals with the aforementioned spectral characteristics. Therefore, the underlying idea behind BYND should find broad resonance in fields where this problem arises, such as engineering, computer science, geophysics, astronomy, or signal processing in general. Thus, experimentalists confronted with challenging signal processing situations will benefit from BYND as well.

Furthermore, while experimentalists may not directly conduct electronic structure calculations themselves, they will definitely benefit from collaboration with theoretical scientists. BYND significantly reduces computational time, making systems more affordable and system sizes more experimentally relevant. We added a note to our revised manuscript in the section titled “Observations, Trends, and Limitations” to further highlight the aforementioned aspects.

It’s worth noting that it is not necessary to use the SMA; any approximate method that yields an initial guess is sufficient. This also means that this approach is not restricted purely to electronic structure applications. As long as the signal is separable into continuum and sparse parts by any kind of initial guess, BYND can yield the correct spectral information. To highlight this aspect, we conducted additional computational simulations of various other systems. These systems include cis-[Ru(4,4'-COOH-2,2'-bpy)₂(NCS)₂] on an anatase (101) cluster as an example of a dye-sensitized solar cell, a molecular ZnPc J-aggregate, a ZnPc film on a Si (111) surface, and zinc-porphyrin molecules on a carbon nanotube (section Observations, Trends, and Limitations).

We have also added to our revised manuscript (section Observations, Trends, and Limitations) a more detailed discussion regarding possible limitations and potential errors. For example, one such source of error could be an insufficient initial guess. Additionally, we provide a more detailed discussion about identifying possible errors effectively without knowing the exact reference spectrum in our supplementary information (section 2).

Reviewer #3 (Remarks to the Author):

This manuscript describes a method for improving the efficiency of calculating spectra for large systems, based on TDDFT. The method seems effective, although I have some comments and concerns. Additionally, the manuscript does not seem like a typical Nature Communications paper. My specific comments are:

- This work is more methodological and detailed than most Nature Communications papers, and it may not have the broad interest required for this journal. However, I did not initially realize that the code is shared (perhaps this would be obvious in the published version but I did not see it in the manuscript). While this does potentially help the broad interest, it is still a detailed, methodological paper.

We have added a code availability statement with a link to download the code. Additionally, we included installation instructions for all required packages.

We acknowledge that our initial draft may have been overly methodological, particularly in the discussion of SMA and RT-TDDFT, which were unnecessarily long and detailed. As a result, we have shortened these sections and lightened up the content. More detailed explanations of these methods are now provided only at the end of the manuscript in the methods section.

- The authors compare computational effort between their method and an “exact” results from 20,000 timesteps to obtain a speedup. However, given that their method also somewhat lowers accuracy, a more fair comparison would be to check how many timesteps it takes to get the same level of accuracy from existing methods. A plot of accuracy vs. CPU time for traditional methods as compared to the new method would be useful. This should be fairly easy to create from the data the authors already have.

With Figure 5 in the new manuscript, we compare the accuracy of BYND with other super-resolution methods such as Compressed-Sensing and Fourier-Padé approximation. We also provide a comparison with the standard Fourier-Transform of a short-time signal. This analysis is based on calculating the Pearson correlation coefficient between the approximated signal and the target signal. By doing so, one quantifies the difference between two spectra as a whole. Therefore, this analysis considers both narrow features and the quasi-continuum. Consequently,

it may lack sensitivity as errors in the continuum may dominate. We provide a more detailed discussion on the limitations of quantifying the differences between two spectra in our methods section.

We conclude that while the Pearson correlation coefficient provides a quantitative measure of similarity, it is often useful to complement this with visual inspection. Therefore, we explicitly decided to plot accuracy vs. the number of time steps to be consistent with Figure 4.

- In Figure 2b, the authors suggest that linear prediction fails when there is a frequency outside of the frequencies predicted by SMA. I do not see concrete evidence that their method solves this particular issue.

With the butadiene molecule we now provide an explicit example for such a case (section 2.2 of our supplementary information).

- Does the alpha value affect the results? Is there any scaling of the raw signal? How will future users choose an appropriate alpha?

Yes, the α value indeed can affect the results and can cause the occurrence of additional narrow features. However, we only observe this at very high α values during our narrow feature optimization. As long as the α value is chosen within a reasonable range (≤ 30), the occurrence of unwanted peaks can be effectively avoided. We demonstrate this in our supplementary information by including an additional analysis on the choice of the ridge regression parameters (section 4).

We also now clearly distinguish between the ridge regression parameter for extracting the sparse signal information (α_{sparse}) and the ridge regression parameter for continuum amplitude determination (α_{cont}). The discussion in our supplementary information also contains suggested default values and ranges of α values to provide general guidance for the user. Intentionally, we show this for a very small number of time steps (1,000) to demonstrate the robustness of our default values (section 4).

All plots have been created without scaling the raw TDDFT dipole signals or the raw signal obtained from BYND. We only scale the SMA signal in Figure 3 so that the maximum amplitude in the time domain matches the maximum amplitude of the TDDFT signal. This is solely for better visualization and allows for a more straightforward comparison between the waveforms of the signals. Therefore, we never provide an R^2 value for the SMA signal. The revised manuscript now includes a note in the caption of Figure 3. We apologize for missing to include this note in the previous version of our manuscript.

- "Provided with enough data points, BYND will yield the exact time dynamics." This seems sensible, but would also be easy for the authors to check to prove it.

We would like to refer to Figure 5 of the new manuscript, which compares BYND with other super-resolution techniques. One can also see that BYND shows a clear convergence behavior.

- "Our method is based on the realization that the spectrum of large systems can be separated in a sparse part and continuum part. " This separation may not always be clear. For example, if a system has several broad features, will the current method be effective?

Broad features that appear as a superposition of many small amplitudes might not be selected in our narrow feature selection process, as we have been using an explicit threshold to select the frequencies from the SMA transition dipole moments. This is exactly the case in systems b) and c) of Figure 6. The broad features at around 7.5 eV were not part of the narrow feature optimization procedure and arise only from the continuum amplitude fitting process, which is done after the narrow features have been assigned. We conclude that as long as the narrow features are correctly reproduced, the continuum fitting procedure is able to accurately yield these broad features as well. A more detailed discussion is provided in our manuscript.

- It is not always easy to see both the black and orange lines, particularly when the orange peaks are larger than the black peaks. Would some adjustment of line thickness/style, translucency, etc. make this easier to see?

As we mentioned in our response to Reviewer 1, we adjusted the transparency for all figures where we explicitly compare our approach with the long-time RT-TDDFT simulation. In the main manuscript, this applies to Figure 4. In the supplementary information, this applies to Figures 4, 15, 16, 17, 18, 19, and 20. We also adjusted the y-range in Figure 4 of our main manuscript. All newly added Figures have also been adapted accordingly.

- The writing is generally fairly good. However, there are a fair number of typos (e.g., duo instead of due). Also, there is a sentence fragment: "Thus, adding only a vanishing amount of computational time."

Thank you for pointing this out. We checked our revised manuscript again for any typos, and we hope that we have found most of them.

Reviewer #3 (Remarks on code availability):

From looking through the repository, it appears to be reasonably user-friendly.

Reviewer #4 (Remarks to the Author):

This manuscript describes a new practical algorithm for computing the electronic absorption spectrum of large systems with many transitions via TDDFT. In short, the method exactly computes a subset of the strongest transitions, and then performs a fitting procedure to approximate the rest. The results are very encouraging. The paper is well written, detailed, and interesting. This work is certain to be of interest to other researchers interested in computing electronic spectra. I recommend publication after the authors have revised the manuscript to address the following questions.

1) Is there any way to know how accurate the spectrum is without comparing to the exact spectrum? That is, could a user estimate the error in their spectrum without computing a better one?

Estimating the error of a spectrum without comparing it to the exact spectrum is quite a challenging task. However, the best strategy would be to compare the error between the target time signal and the one obtained from BYND. In other words, one would determine the goodness of our fit by using methods such as mean absolute error (MAE), mean squared error (MSE), root mean squared error (RMSE), or the coefficient of determination (R^2). One would then either try to minimize the error or maximize R^2 .

Regardless of the chosen method, the cyclic nature of the data set poses some limitations on such an analysis. Strictly speaking, one cannot compare goodness-of-fit values between time series of different lengths. Some parts of the signal might be reproduced better than others, thus calculating the goodness of fit for one signal length might yield a different value compared to another signal that was cut differently. We also want to emphasize that in the case of BYND, only the values calculated between the narrow feature signal and the target signal are meaningful. The fitting of continuum amplitudes is an under-determined system that will always reproduce the correct signal in the end.

Despite these limitations, performing a goodness-of-fit analysis is still extremely useful for analyzing the error of the calculated spectrum. We provide examples in our supplementary information where we use the R^2 value to spot errors associated with the SMA input. Such cases can include situations where not enough frequencies have been selected or where a too-small $\Delta\omega_{\text{init}}$ was applied. In general, we find that the R^2 value is quite helpful in estimating the error with respect to the chosen input parameters. Thus, R^2 could be used as a score value to judge the outcome of the narrow feature optimization (see supplementary information section 2).

2) Similarly, it would be interesting to quantify the convergence of the spectra in figure 4 with respect to the propagation time. Does the error decrease predictably with increasing simulation time, and if so, to what order does the error scale?

With Figure 5 in our revised manuscript, we now show the convergence with respect to propagation time by quantifying the error using the Pearson correlation coefficients. We clearly see that BYND predictably converges towards the exact result of our 20,000 time step reference. From this analysis, we also see that the error decreases following the power law, 10^{kx} , with x being the number of time steps and k being a negative number. In our particular example,

we utilize our two largest nanocrystal systems, $\text{Cd}_{33}\text{Se}_{33}/\text{Zn}_{93}\text{S}_{93}\text{-2}(\text{ZnPc})$ and $\text{Cd}_{33}\text{Se}_{33}/\text{Zn}_{93}\text{S}_{93}\text{-2}(\text{ZnPc})\text{-DPA}$, and average over the obtained correlation coefficients.

3) Will SMA necessarily provide a good prediction of the number of narrow features? For example, plasmonic excitations are collective motions of many electrons, and therefore typically exist as superpositions of many amplitudes. Would the approach describe in this work be able to handle excitations like this?

A superposition of many small amplitudes might be missed in our narrow feature selection process. The transition dipole moments of each single frequency are probably too low compared to other, more "bright" features to be selected by a given threshold. However, with systems b) and c) in Figure 6 of our revised manuscript, we provide an example of a situation where this happens. The frequencies that contribute to the broad feature at around 7.5 eV were not part of the narrow feature optimization procedure. However, these features arise from adjusting the continuum amplitudes. This demonstrates that as long as the narrow features are correctly assigned, the continuum amplitude fitting procedure is able to give an accurate prediction of very broad features as well. Our newly revised manuscript contains a discussion of these situations.

In general, the quality of the outcome will depend on how good the SMA prediction is. We already mentioned a few limitations in our answer to point 1), but we would like to highlight that one does not necessarily need to rely on the SMA. Any inexpensive approximated method can be used to provide initial guess frequencies. To name just a few, such methods could be time-dependent tight-binding methods, time-dependent DFTB, or time-dependent INDO/S. Also this is mentioned in our revised manuscript (section Observations, Trends and Limitations).

Reviewer #1 (Remarks to the Author):

The authors have addressed all my comments. I therefore recommend this manuscript for publication in Nature Communications as is.

Reviewer #2 (Remarks to the Author):

I recommend the revised manuscript for publication, since the concerns of the reviewers are satisfactorily addressed in sufficient detail and the changes to the revised manuscript seem appropriate.

Reviewer #3 (Remarks to the Author):

The authors have put significant effort into the revisions, which have significantly improved the work. They have addressed my scientific concerns to my satisfaction.

Reviewer #4 (Remarks to the Author):

My comments have been addressed very thoroughly. I recommend publication of this manuscript in its current form.